

# Effects of early life stage exposure of largemouth bass to atrazine or a model estrogen (17α-ethinylestradiol)

Jessica K. Leet[1], Catherine A. Richter[1], Robert S. Cornman[2],
Jason P. Berninger[1], Ramji K. Bhandari[3], Diane K. Nicks[1,†]
James L. Zajicek[1], Vicki S. Blazer[4] and Donald E. Tillitt[1]

[1] Columbia Environmental Research Center, United States Geological Survey, Columbia, MO, USA
[2] Fort Collins Science Center, United States Geological Survey, Fort Collins, CO, USA
[3] Department of Biology, University of North Carolina at Greensboro, Greensboro, NC, USA
[4] Leetown Science Center, United States Geological Survey, Kearneysville, WV, USA
[†] Deceased author.

Corresponding author
Jessica K. Leet, jleet@usgs.gov

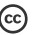

## ABSTRACT

Endocrine disrupting contaminants are of continuing concern for potentially contributing to reproductive dysfunction in largemouth and smallmouth bass in the Chesapeake Bay watershed (CBW) and elsewhere. Exposures to atrazine (ATR) have been hypothesized to have estrogenic effects on vertebrate endocrine systems. The incidence of intersex in male smallmouth bass from some regions of CBW has been correlated with ATR concentrations in water. Fish early life stages may be particularly vulnerable to ATR exposure in agricultural areas, as a spring influx of pesticides coincides with spawning and early development. Our objectives were to investigate the effects of early life stage exposure to ATR or the model estrogen 17α-ethinylestradiol (EE2) on sexual differentiation and gene expression in gonad tissue. We exposed newly hatched largemouth bass (LMB, *Micropterus salmoides*) from 7 to 80 days post-spawn to nominal concentrations of 1, 10, or 100 μg ATR/L or 1 or 10 ng EE2/L and monitored histological development and transcriptomic changes in gonad tissue. We observed a nearly 100% female sex ratio in LMB exposed to EE2 at 10 ng/L, presumably due to sex reversal of males. Many gonad genes were differentially expressed between sexes. Multidimensional scaling revealed clustering by gene expression of the 1 ng EE2/L and 100 μg ATR/L-treated male fish. Some pathways responsive to EE2 exposure were not sex-specific. We observed differential expression in male gonad in LMB exposed to EE2 at 1 ng/L of several genes involved in reproductive development and function, including *star*, *cyp11a2*, *ddx4* (previously *vasa*), *wnt5b*, *cyp1a* and *samhd1*. Expression of *star*, *cyp11a2* and *cyp1a* in males was also responsive to ATR exposure. Overall, our results confirm that early development is a sensitive window for estrogenic endocrine disruption in LMB and are consistent with the hypothesis that ATR exposure induces some estrogenic responses in the developing gonad. However, ATR-specific and EE2-specific responses were also observed.

## INTRODUCTION

Early life stages in fish and other vertebrates tend to be sensitive to effects of endocrine disrupting compounds (EDCs) (*Van Aerle et al., 2002*), and EDCs are considered a global concern. Effects during gonad development are likely to be permanent, organizational events with potential for complete sex reversal, in contrast to the generally reversible, activational effects that occur during adulthood. Fish early life stages are of particular concern in agricultural areas, as a spring influx of pesticides coincides with spawning and early development (*Gall et al., 2011*).

Evidence of endocrine disruption in wild fish has been observed in many areas. A high prevalence of testicular oocytes, an intersex condition, has been observed in smallmouth bass within the Potomac (*Blazer et al., 2007*, *2010*; *Iwanowicz et al., 2009*) and Susquehanna (*Blazer et al., 2014*) river basins of the Chesapeake Bay watershed (CBW) in the eastern United States. Throughout North America, intersex gonads have been found in black bass species including smallmouth bass (*Micropterus dolomieu*) and largemouth bass (*Micropterus salmoides*) (*Abdel-Moneim et al., 2015*; *Blazer et al., 2018*; *Grieshaber et al., 2018*; *Hinck et al., 2009*; *Iwanowicz et al., 2016*; *Kellock et al., 2014*; *Yonkos, Friedel & Fisher, 2014*). The individual chemicals, complex mixtures of chemical compounds and/or other environmental stressors contributing to the development of intersex in specific areas are not yet fully understood. A significant positive correlation was observed between the incidence of intersex and the concentration of Atrazine (ATR) in the water during monitoring of the Potomac river basin at six sites in Maryland, Virginia and West Virginia during the spring (*Kolpin et al., 2013*). This observation highlighted ATR as a contaminant of interest for further investigation into its potential role as an estrogenic endocrine disrupting contaminant in the CBW.

Atrazine has been implicated in both reproductive dysfunction in adults and alterations during early development in fish, as well as in other vertebrates (*Wirbisky & Freeman, 2015*). Adult exposures to atrazine (ATR) impaired reproduction in medaka (*Oryzias latipes*) (*Papoulias et al., 2014*) and fathead minnow (*Pimephales promelas*) (*Tillitt et al., 2010*). However, the molecular mechanism of endocrine disruption by atrazine remains unclear and appears distinct from the mechanism of estrogenic EDCs (*Richter et al., 2016*). Atrazine does not directly activate the estrogen receptor, but has been hypothesized to induce expression of *cytochrome P450, family 19, subfamily A, polypeptide 1a* (*cyp19a1a*, previously *aromatase*) through inhibition of phosphodiesterase; activation of nuclear receptor subfamily five, group A, member 1a (Nr5a1a); or alterations in miRNA expression (*Sanderson et al., 2001*; *Roberge, Hakk & Larsen, 2004*; *Suzawa & Ingraham, 2008*; *Wang et al., 2019*). Early life stage ATR have been reported to alter sex ratios (*Suzawa & Ingraham, 2008*; *Wang et al., 2019*).

Exposure to estrogenic compounds, including the pharmaceutical estrogen 17α-ethinylestradiol (EE2), has been shown to induce intersex in certain fish species (*Abdelmoneim et al., 2019*). There have been many studies evaluating the effects of EE2 on various fish species at different life stages, especially the model species zebrafish (*Danio rerio*), medaka, and fathead minnow. However, the exact mechanism causing development of intersex gonads has not been identified and there is a particular data gap in this area for black bass. This early life stage evaluation has not been done in a laboratory setting in black bass species, which are economically important sport fish. Identifying estrogen-specific responses in early life stage largemouth bass can provide a baseline to build on when identifying mechanisms of action, as estrogens have been shown to have additive effects (*Brian et al., 2005*).

To evaluate the role of atrazine as an EDC and the role of estrogenic EDCs in intersex, largemouth bass fry were exposed to a range of concentrations of ATR or EE2 during sexual differentiation. To our knowledge this study is the first to measure effects of laboratory ATR and EE2 exposure in early life stage LMB. The objectives of this study were to: monitor the growth of LMB and development of the gonads through histological examinations; identify differences in gene regulation associated with ATR and EE2 exposure; and develop testable hypotheses for potential biochemical pathways and cellular mechanisms leading to altered gonad development, the condition of intersex, and ultimately impairment of reproductive function in largemouth bass. The focus of this manuscript is on potential molecular initiating pathways that could lead to altered gonad development and function.

# MATERIALS AND METHODS

## Experimental design

Largemouth bass fry were exposed to solvent control (0.0001% ethanol), 17α-ethinylestradiol (EE2, 1 or 10 ng/L as a model estrogen control), or atrazine (ATR, 1, 10, or 100 μg/L) starting at 7 days post spawn (dps). Exposures continued to 80 dps, during early gonadal and sexual differentiation. Each treatment was conducted in quadruplicate ($n = 4$), for a total of 24 exposure tanks (6 treatments × 4 replicates). Fry/juveniles were sampled at 17, 33 and 80 dps. Each tank began with approximately 500 viable fry at 7 dps. At the first two time points, 20 fish per tank were collected for histological examination of gonadal development. At 80 dps 140 fish per treatment (35 fish per tank) were sampled to assess growth, sex identification. Of those fish sampled at 80 dps, gonads of 20 fish per treatment (five fish per tank) were removed for histological sex identification and gene expression analysis. Four treatments were chosen for gene expression analysis: solvent control, low EE2 (1 ng/L) and low and high ATR (1 and 100 μg/L). Four fish were randomly selected per treatment per sex (4 fish × 4 treatments × 2 sexes) for a total of 32 samples analyzed by RNAseq.

All exposure, sample processing, and data analysis was conducted at the U.S. Geological Survey Columbia Environmental Research Center (CERC, Columbia, MO, USA) unless otherwise stated. This study was in compliance with all applicable sections of the Final Rules of the Animal Welfare Act regulations (nine CFR) and with all CERC Institutional

Animal Care and Use Committee guidelines for the humane treatment of the test organisms during culture and experimentation.

## Spawning, egg collection, feeding and animal care

The original broodstock of virgin largemouth bass were obtained from the U.S. Fish and Wildlife Service's Genoa National Fish Hatchery, Genoa, WI. Eggs for the current study were collected from second generation broodstock reared at CERC. Approximately 12–18 sexually mature largemouth bass (6–9 from each sex) were spawned in a 0.2 acre pond containing 45 cm × 80 cm spawning mats placed in shallow near-shore water. Mats with freshly deposited eggs were collected daily, brought into the laboratory and treated with sodium sulfite (0.015% solution) for 3–5 min to release the eggs from the mats. The eggs were then placed in modified MacDonald egg incubation tubes and rolled at a temperature no more than 1 °C from the pond temperature. Eggs were collected until the total was greater than 24,000 eggs to allow for 1,000 embryos per tank (total of 24 tanks). Eggs were rolled with a stream of fresh well water in the egg incubation rack until hatch. Hatched fry were collected in mixed cultures and maintained at temperature (22–24 °C) until placed in the exposure tanks. Upon placement in the exposure tanks, the fry were acclimated to 22 °C and maintained there for the remainder of the study period under a 16:8 light:dark photoperiod.

Hatched fry were fed a combination of three different food sources over the course of the study. Live freshwater rotifers (*Brachionus calyciflorus*) were fed in conjunction with live, newly hatched brine shrimp (*Artemia* sp.) nauplii to satiation three times a day during the first week after initiation of exogenous feeding. After the first week, fish were transitioned off rotifers and offered a combination of live, newly hatched *Artemia* nauplii three times a day and a dry manufactured diet (Otohime, Reed Mariculture, Campbell, California, USA) to satiation twice a day. The manufactured diet was administered to the fish via an automatic feeder to each tank. The manufactured diet was adjusted throughout the study based on growth rates and gape width. Tanks were siphoned clean once every other day to remove uneaten food and waste. Additionally, fish were prophylactically treated in-tank twice a week with 20 parts per million (ppm) Chloramine T for the first four weeks followed by 10 ppm twice a week for the remainder of the study to prevent bacterial outbreaks from occurring.

Well water was used in this study and conditions were maintained within criteria set forth by ASTM International (*American Society for Testing & Materials (ASTM), 2004*) for toxicity testing with aquatic organisms. General water quality was expected to be 275 mg/L hardness, 245 mg/L alkalinity, 8.3 pH, 0.0200 mg/L ammonia. Alkalinity, hardness, pH, gas saturation and ammonia were monitored weekly. Dissolved oxygen and temperature were measured every other day. Tanks were rectangular glass aquaria with 64 L capacity, filled to 48 L. Each tank had an overflow outlet into the waterbath and an airstone. The sides of each tank were covered with contact paper to prevent visual interactions of fish tank to tank. Volume of water flowing into the tanks was checked weekly.

## Exposure and chemical analysis

Exposures were conducted in proportional, flow-through diluters with approximately four tank turnovers per day. Exposure chemical concentrations were confirmed twice weekly for the first two weeks, then weekly for the remainder of the exposures. Atrazine (98% purity, Fluka, College Park, GA, USA) was prepared in stock solutions of 25 mg/L in CERC well water and stored in amber bottles at 4 °C prior to use. 17α-Ethinylestradiol (Sigma–Aldrich, St. Louis, MO, USA) was prepared in a stock solution of 10 μg/mL in ethanol. Working solutions in CERC well water of ATR were prepared for final exposure concentrations of 0, 1, or 100 ug/L and of EE2 for final exposure concentrations of 1 or 10 ng/L and solvent concentration of 0.0001% ethanol. Diluters were intermittent-flow exposure systems with Hamilton syringes delivering the test chemicals to replicate exposure chambers. System cycling occurred at a rate of six cycles per hour. The system was equilibrated with the test chemicals for 5 days prior to stocking the fish. At the commencement of the exposure, newly hatched fry (7 dps) were placed in floating baskets within randomly assigned aquaria for each treatment/replicate combination. Atrazine in the tank water was quantified using enzyme-linked immunosorbant assay (ELISA) kits (Abraxis, Warminister, PA, USA) in accordance with manufacturer's protocols. Confirmatory analysis was performed on selected water samples by gas chromatography (*Jiménez et al., 1997*). Briefly, water samples were extracted using methylene chloride; the extract dried with sodium sulfate and filtered through glass fibers; volume reduced to 0.1 mL in methyl tertiary butyl ether; and triphenylphosphate (Chem Service Inc., West Chester, PA, USA, 500 μg/mL in MtBE) was added as an instrumental internal standard. The extracts were analyzed by gas chromatographic nitrogen/phosphorus detector (GC/NPD) and quantified by Perkin-Elmers TotalChrom™ workstation chromatography data software. Quality control samples were analyzed with each sample set and included: ATR-spiked water, matrix (well) water blank, and a procedural blank. Concentrations of EE2 in exposure tank water were monitored using an ELISA kit by Ecologiena, (Tokiwa Chemical Industries Co. Ltd, Japan) according to manufacturer's instructions. Water samples were brought to pH 7.0 prior to filtering through a glass fiber filter. Samples were then extracted and concentrated using C18-solid phase extraction. Extracted samples were added to a 96-well microtiter plate that was coated with polyclonal rabbit anti-EE2 antibodies. Following incubation, a tracer conjugated with horseradish peroxidase was applied. After tracer incubation the plate underwent washing and the addition of a color substrate (3,3′,5,5′-tetramethylbenzidine, TMB). After color development a stop solution (sulfuric acid) was added before reading the absorbance at 450 nm. The EE2 concentrations were determined by quantifying the absorbance values in relation to the measured values of EE2 calibration standards that had been assayed in the same manner.

## Fish collection

Fish were euthanized with an overdose (300 mg/L) of MS-222 (Ethyl 3-aminobenzoate methanesulfonate, Sigma–Aldrich, St. Louis, MO, USA), blotted dry, then weight and total length were measured. Midsections were sampled from 30 fish per treatment; after

removing the head just posterior to the opercular flap and removing the tail just posterior to the anal pore, midsections were preserved (see below) for histological analysis.

Fish sampled for gene expression analysis had an incision made from anal pore to opercular flap, and a panel of muscle tissue removed to obtain access to the juveniles' gonads. RNAlater (Sigma–Aldrich, St. Louis, MO, USA) was used to immediately preserve the gonads still in the body cavity, and to increase visualization and integrity of the gonad tissue. The gonad was then carefully removed and divided. One lobe was preserved (see below) for histological analysis and sex identification. The second lobe was stored in RNAlater at 4 °C overnight, then transferred to −20 °C until RNA was extracted.

## Histological analysis

Two different histological samples were taken at 80 dps: (1) whole, intact midsections or (2) individual gonad lobes were extracted. Samples were preserved in Z-fix (Anatech Ltd, Battle Creek, MI, USA) or PAXgene tissue fix (PreAnalytix, Hombrechtikon, Switzerland) for 12–16 h then moved to 70% ethanol and processed within 3 weeks. Midsections were cut in the center and positioned in the paraffin block so both anterior and posterior directions of the midsection could be visualized. Samples were processed by dehydration and embedded into paraffin (*Luna, 1968*). The midsections were sectioned in 5 µm transverse cuts until the gonads were clearly visible. Kidney and swim bladder were used to orient the visualization of the gonad, as the gonads are on the ventral side of the swim bladder in the section of the fish where the hind kidney is clearly visible. Slides were stained with hematoxylin and eosin (*Luna, 1968*). In those fish where gonads were extracted during sampling, RNAlater (ThermoFisher, Waltham, MA, USA) was added to the open cavity of the fish to give the gonads integrity to be more easily removed. Then gonads were removed with forceps, one lobe was taken for histology and the other was preserved in RNAlater for RNAseq (five fish per treatment). Individual lobe samples collected for histological analysis were stored in Paxgene for 12–16 h and then transferred to 70% ethanol and processed within 3 weeks. For each individual gonad lobe approximately 20–40 sagittal 5 µm thick sections were taken.

The characteristics used to identify female fish were: gonadal tissue had the presence of primary oocytes (POs; round cells with hematoxylin staining around the nucleus that was lighter or stained with eosin, and larger than 25 µm in diameter), ovarian tissue was relatively larger than testes (larger than 100 µm in diameter) and round in shape, typically with an ovarian cavity present. Fry were identified as female if POs were present in any of the sections. A presumptive male identification was made if only primordial germ cells (PGCs) and no POs were present in the gonad sections. It is assumed these undifferentiated gonads were male because this concurs with expected development of male LMB (*Johnston, 1951*). Due to the lack of testis differentiation at this developmental time point, there is a possibility that these undifferentiated gonads could be ovaries with considerably delayed development. However, due to the likelihood that these were males, those identified as presumptive males will hereafter be referred to as males. In females the extreme anterior and posterior ends of ovaries appeared similar to undifferentiated gonadal tissue, except with no PGCs present. When undifferentiated

tissue with no PGCs was sectioned further, POs were consistently found, leading to a female identification. Conversely, when undifferentiated gonadal tissue with any PGCs was sectioned through the entire gonad, no POs were revealed, leading to a male identification. If only connective tissue was present and no gonadal tissue could be identified, sex identification was deemed inconclusive for that fish.

## RNA extraction, library preparation and sequencing

Total RNA was extracted from an individual gonad lobe from each of the selected fish, corresponding to a gonad lobe that was processed for histological sex identification. The extraction was performed using an RNeasy Mini Kit (74104; Qiagen, Hilden, Germany). Briefly, tissue was removed from RNAlater and homogenized in 150 µl of lysis buffer using a micro-tube homogenizer and pestle (Fisher). The pestle was rinsed with an additional 200 µl of buffer following homogenization (350 µl total) to assure no sample was lost. Extraction was performed according to manufacturer's instruction, with DNase treatment (Qiagen, Hilden, Germany). RNA was eluted off the extraction column in two 14 µl elutions (28 µl final volume). Capillary electrophoresis on a QIAXEL instrument (Qiagen, Hilden, Germany) was used to characterize the concentration and quality of the RNA following manufacturer's instructions. Four samples per treatment were chosen for the RNAseq analysis. These samples had RNA integrity scores (RIS) with an average of 7.73 +/− 0.97. Concentrations of RNA were normalized for all samples. Library preparation and sequencing was performed by the DNA Core facility at the University of Missouri (Columbia, MI, USA). Libraries were prepared by Ultra Low RNA Library Preparation kit (Clontech, Mountain View, CA, USA) following manufacturer's instructions. Transcripts were sequenced in four lanes of a 2 × 75 NextSeq (Illumina, San Diego, CA, USA) paired-end read.

## Assembly of gonad transcriptome

Initial read processing and assembly into transcript contigs was performed by the sequencing core. The reads were first trimmed of low-quality base calls and adapter sequence, if present, by the Trimmomatic command (*Bolger, Lohse & Usadel, 2014*) in Trinity (Trinity-v2.3.2) with default settings except the five bases were not removed from each end. Strand specific paired end settings were used assuming the first read was on the sense (forward) strand and the second read was on the antisense (reverse) strand. All settings were default including the k-mer size of 25. The initial assembly included 743,843 contigs, which was reduced in number to 103,580 by requiring at least 10 counts summed across all samples and imposing a threshold contig length >200 nt.

Prior to differential expression analysis, contigs were clustered at 95% with cd-hit-est (*Li & Godzik, 2006*) to further reduce redundancy in the reference assembly, to 59,261 contigs (N50 of 3,545 nt). Assembled and clustered RNA contigs will be referred to as transcripts hereafter. Transcripts were annotated using a BlastX search against all *Danio rerio* peptides (version GRCz10, NCBI accession number GCF_000002035). A reciprocal search identified 11,916 one-to-one matches at a minimum bit score of 100, for which functional annotations were used in enrichment analysis (see below).

Annotated transcripts will be referred to as gene transcripts or expressed genes. The average number of mapped reads was 47.4 +/− 7.4 million reads/sample.

## Analysis of differential transcript expression

For quantification of transcripts, raw reads were reprocessed using a different workflow than that used for assembly. Reads were trimmed with CLC Genomics Workbench v 9.5 (Qiagen, Hilden, Germany) at an error probability of 0.05, allowing a maximum of two ambiguous bases and requiring a minimum trimmed length of 50 bp. Sequencing adapters were removed using default scoring parameters. Abundance was estimated with kallisto (*Bray et al., 2016*) using a k-mer size of 31 and with sequence composition bias correction. Note that counts estimated by kallisto are probabilistically adjusted to account for sequence shared among multiple transcripts (as in close paralogs or alternative isoforms), resulting in fractional count values. Counts were rounded to integer values for analysis with edgeR (*Robinson, McCarthy & Smyth, 2009*).

The multidimensional scaling function of edgeR was used to check for consistency of response by treatment. Two samples (2–2 and 5–5) were strong outliers by this approach and therefore removed from all further analyses. For each contrast tested, low-abundance transcripts were filtered by requiring at least 2 cpm in at least three samples, unless the sample size had been reduced due to the exclusions noted above, in which case at least two samples with ≥2 cpm were required. The *p*-value for each transcript was calculated using the "exactTest" function in edgeR, based on the normalized and expression-filtered data for each pairwise contrast and adjusted using the Benjamini–Hochberg correction (*Benjamini & Hochberg, 1995*) to account for false discovery and an adjusted value of 0.05 was considered the threshold of significant differential expression. No minimum fold-change threshold was used. Genes with significantly male-biased and female-biased expression were identified by comparing male and female expression in the controls using the same thresholds.

## Functional enrichment analysis

Three differentially expressed (DE) gene sets of interest were further analyzed in this study: (1) female-biased DE genes (female expression significantly greater than male) that were also differentially expressed in response to 100 μg ATR/L and 1 ng 17α-ethinylestradiol/L treatments in males; (2) DE genes in common among both 1 and 100 μg/L ATR treatments in males; and (3) DE genes in common among both 1 and 100 μg/L ATR treatments in females. For each comparison (gene sets 1, 2 and 3; see section "Gene Ontology Analysis"), enrichment of ontologies and pathway associations among DE genes were evaluated with go-seq (*Young et al., 2010*), using only the annotations associated with transcripts with one-to-one matches to *D. rerio*. Go-Slim gene ontology annotations for the parent Ensembl (*Hubbard et al., 2002*) gene ID of each matched *D. rerio* protein were downloaded from BioMart (*Smedley et al., 2009*) on 10 January 2017. NCBI gene IDs corresponding to each Ensembl gene were downloaded from the public ftp site on 18 January 2017.

## Statistical analysis

Statistical analysis of physiological endpoints was performed using JMP 13.1.0 (SAS, Cary, NC, USA) with significance set at 5% ($p < 0.05$) for all comparisons. Survival, weight and length were analyzed using analysis of variance (ANOVA) followed by Dunnett's post hoc test to identify treatments that differed significantly from the solvent control. Sex ratios were compared across treatments using a chi-square test.

# RESULTS

## Exposure, mortality, growth and sex identification

### Chemical exposure

Concentrations of ATR during the course of the exposure averaged 0.91 (±0.06), 9.87 (±0.68), and 105.09 (±8.57) μg ATR/L corresponding to nominal concentrations of 1, 10 and 100 μg ATR/L, respectively. Concentrations of EE2 averaged 0.84 (±0.04) and 7.34 (±0.46) ng EE2/L for corresponding nominal concentrations of 1 and 10 ng EE2/L, respectively. The concentrations in all treatments remained stable throughout the study period (Fig. 1).

Mortality over the 73 d study ranged from 52.9% to 58.4% among tanks in all treatment groups, and there were no significant differences between treatments (Table 1; Table S1). High mortality is expected during early development. We also observed cannibalism among the test organisms that limited survival, consistent with the ecological role of LMB as voracious piscivores from early in development.

### Gonad development and sex identification

Developing gonads in midsections at 17 dps were small and had connective tissue and primordial germ cells (PGCs) (Figs. 2A and 2B). At 33 dps the gonads were slightly larger and PGCs were observed (Figs. 2C and 2D). At 80 dps testis tissue remained mainly undifferentiated with the presence of clear PGCs, but no testis-specific characteristics (Figs. 2E and 2F). Ovarian tissue began to differentiate by 80 dps, exhibiting ovarian cavities and primary oocytes in addition to PGCs, and ovaries were much larger than testes in cross-section at this timepoint (Figs. 2G and 2H).

Equal effort was made for sex identification across tanks; the reported n of each sex for each treatment reflects confident sex identification (Table 1; Table S1). To verify consistency in the sex identification protocol, 70% of samples had one or two additional blind identifications performed, of which 100% concurred with the original sex identification. The only significant difference in sex ratio from the expected 1:1 female: male was the 10 ng EE2/L treatment, where all fish in each replicate were identified as female, except a single individual from one replicate identified as a male.

### Juvenile growth

There were no significant differences in length or weight between treatment groups relative to control when averaged across both sexes. When sex-specific responses were analyzed, female length was significantly greater in the both the 1 and 10 ng EE2/L treatments compared to control (Table 1; Table S1). Female weight was significantly increased in the

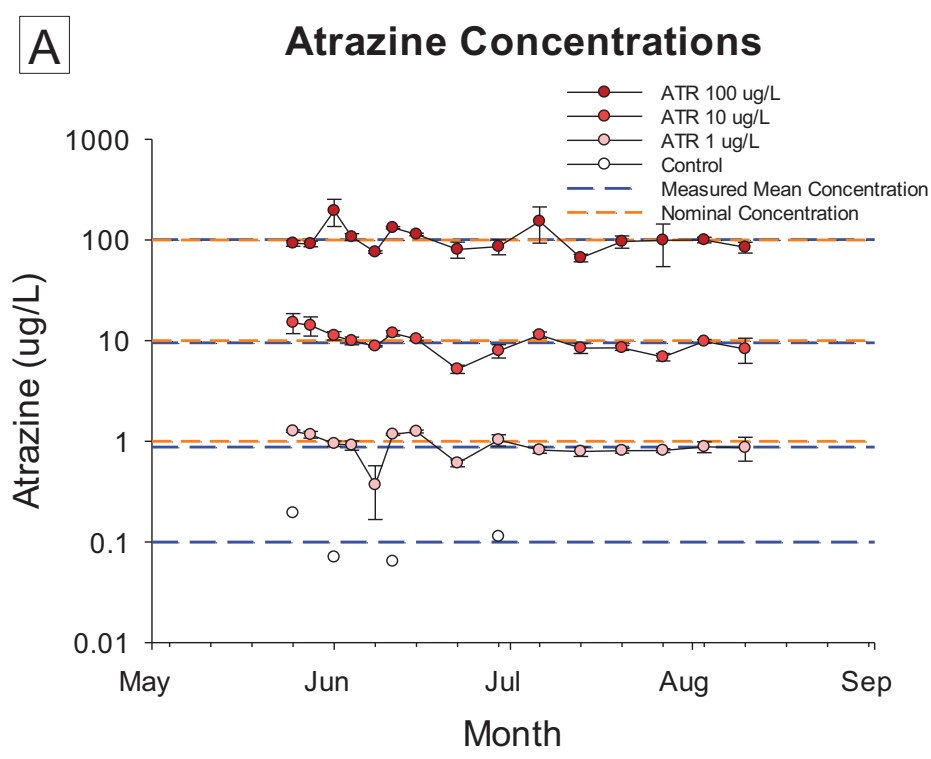

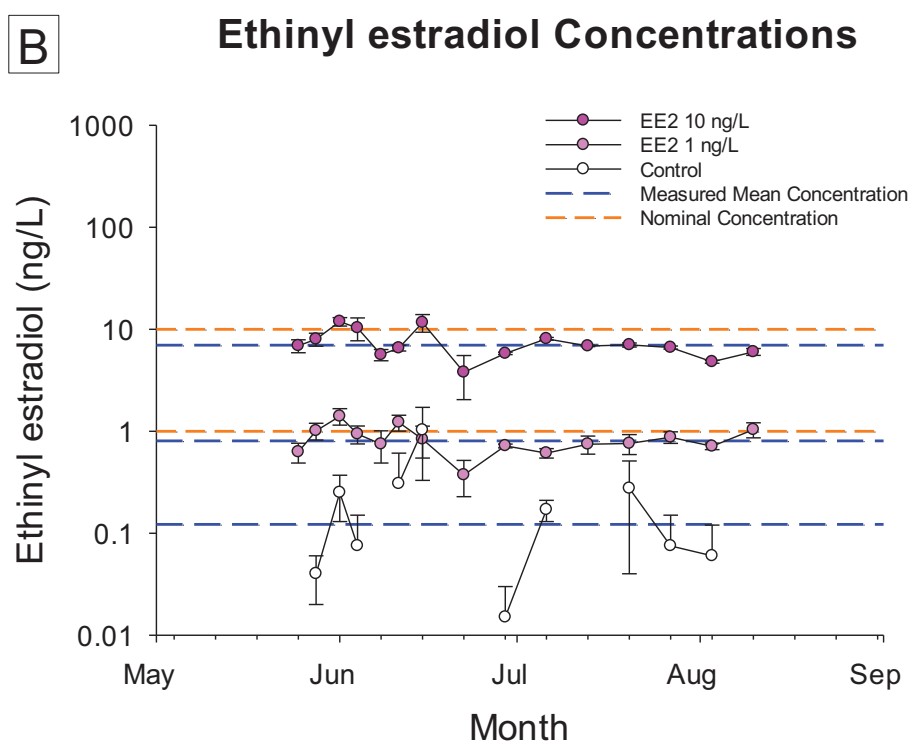

**Figure 1 Chemical concentration data.** Concentrations of atrazine (ATR, (A)) and 17α-ethinylestradiol (EE2, (B)) at various time points during the 70 day flow-through exposure. All replicates measured at each time ($n = 4$) for all treatments, except for the solvent control ($n = 1$ for ATR, $n = 2$ for EE2). Error bars represent standard deviation.

**Table 1 Summary of growth and sex ratio data.** Summary of Mean (SE) of growth and sex ratio data measured in 80 dps largemouth bass juveniles that were successfully identified as either female or male.

| Treatment | Survival (%) | n | | Length (mm) | | Weight (g) | | Sex ratio |
|---|---|---|---|---|---|---|---|---|
| | | Female | Male | Female | Male | Female | Male | % Female |
| Control | 58.4 (2.6) | 57 | 52 | 54.2 (1.0) | 54.1 (1.4) | 1.70 (0.16) | 1.62 (0.15) | 52.5 (5.3) |
| 1 ng/L EE2 | 55.6 (2.2) | 42 | 61 | 60.3 (0.4)* | 58.3 (1.4) | 2.37 (0.15)* | 1.95 (0.11) | 40.3 (6.4) |
| 10 ng/L EE2 | 52.9 (1.4) | 94 | 1 | 59.5 (2.3)* | 64.0 | 1.98 (0.22) | 1.93 | 99.0 (1.0)**†† |
| 1 µg/L ATR | 54.0 (2.9) | 44 | 59 | 55.8 (0.6) | 57.1 (1.5) | 1.68 (0.09) | 1.48 (0.13) | 42.7 (3.5) |
| 10 µg/L ATR | 58.2 (2.3) | 38 | 56 | 58.8 (1.3) | 56.8 (2.3) | 2.15 (0.18) | 1.80 (0.23) | 39.7 (5.4) |
| 100 µg/L ATR | 54.1 (2.3) | 55 | 58 | 58.7 (1.1) | 60.1 (1.2) | 2.08 (0.17) | 2.23 (0.16) | 48.4 (4.6) |

Notes:
* $p < 0.05$ for treatment means significantly different from control tested by Dunnett's post hoc.
** $p < 0.01$ for treatment means significantly different from control tested by Dunnett's post hoc.
†† $p < 0.01$ for sex ratios significantly different than expected 1:1 female:male tested by chi-square analysis.

1 ng EE2/L treatment group, but not the 10 ng/L treatment group, compared to control (Table 1; Table S1). There were no significant differences between treatment groups for length or weight in males.

## Gene expression analysis
### Differential expression
RNAseq analysis identified 41,565 total transcripts and 28,314 total DE transcripts among all comparisons (BioProject Accession PRJNA485177). Multidimensional scaling of DE transcripts revealed clustering of expression patterns in the 1 ng EE2/L and 100 µg ATR/L treated male fish (Fig. 3). In females, expression patterns in both atrazine treatments clustered with the controls. Many transcripts were significantly differentially expressed between sexes in the controls, with slightly more male-biased (expression significantly greater in males) than female-biased (expression significantly greater in females; Fig. 4). Overall, males exhibited more DE transcripts in response to treatments than females (Fig. 4). Few DE transcripts were common between ATR-treated males and females. Most responsive transcripts in males were down-regulated. In females, most responsive transcripts were upregulated, particularly male-biased transcripts (M>F; Fig. 4). In the 1 ng/L EE2-exposed fish, of those transcripts differentially expressed in gonad (3,376 in males, 1,931 in females), 721 were identified in both sexes. There was a strong correlation of log-fold changes between males and females among these 721 transcripts ($R^2 = 0.92$).

### Gene Ontology analysis
We developed three gene sets of interest based on intersections among gene sets responsive to different conditions. To test the hypothesis that atrazine exposure had estrogenic effects on gene expression in male gonad, we examined gene set 1: female-biased DE genes (female expression greater than male in controls) that were also differentially expressed in response to both the 100 µg ATR/L and 1 ng EE2/L treatments in males. We predicted that more genes with higher expression in ovary would be upregulated in response to both 1 ng EE2/L and 100 µg ATR/L exposures in males. To investigate atrazine-specific gene

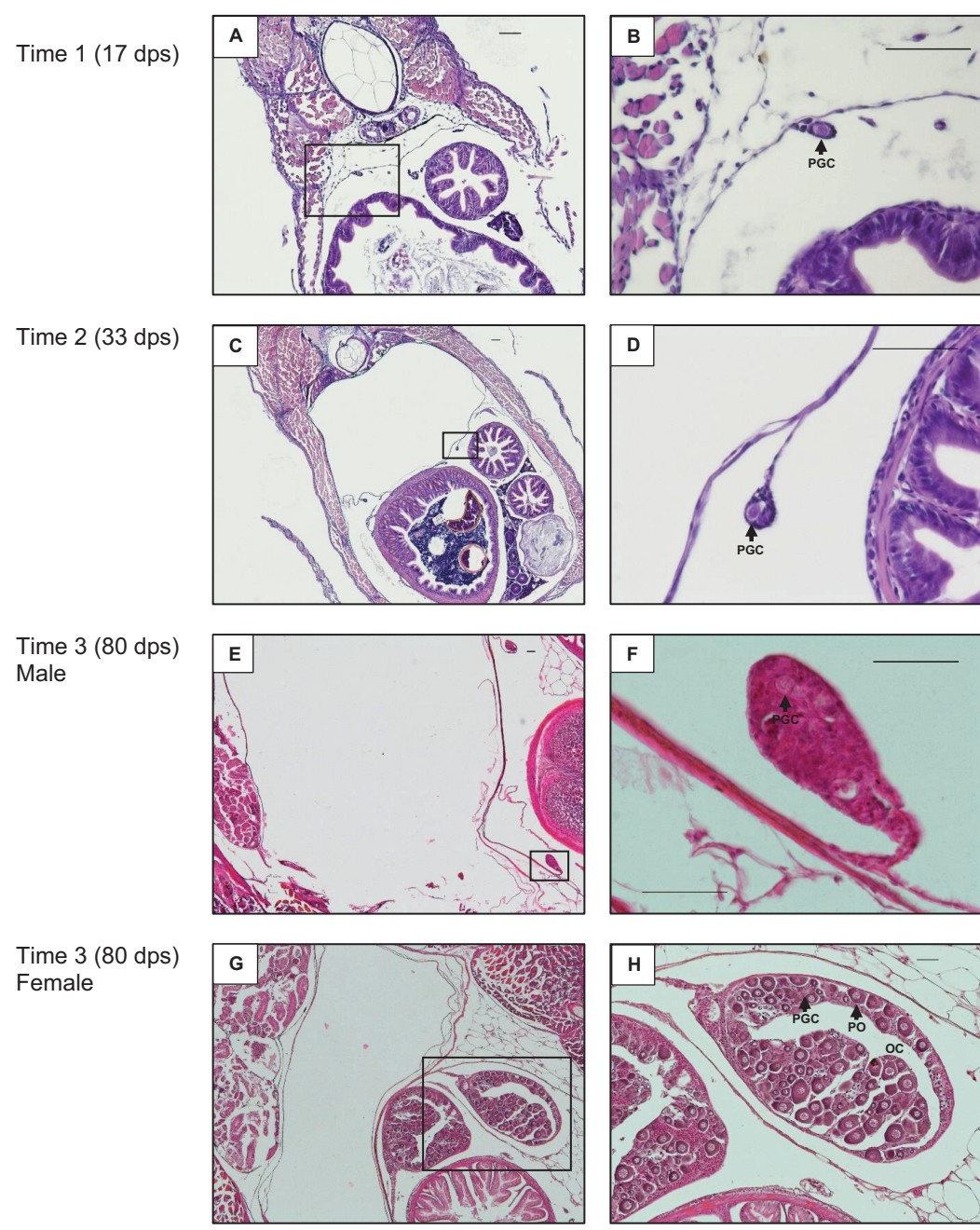

Time 1 (17 dps)

Time 2 (33 dps)

Time 3 (80 dps)
Male

Time 3 (80 dps)
Female

**Figure 2 Gonad development.** Example of largemouth bass gonad development from the control treatment at 17 days post-spawn (dps) at 100× (A) and 400× (B) magnification, 33 dps at 40× (C) and 400× (D) magnification, 80 dps male at 40× (E) and 400× (F) magnification, and 80 dps female at 40× (G) and 100× (H) magnification. The bar in the upper right of each panel equals 50 μm. PGC, primordial germ cell; PO, primary oocyte; OC, ovarian cavity.

expression fingerprints in males, we examined gene set 2: DE genes in common among ATR treatments at both 1 and 100 μg/L in males. Finally, to investigate atrazine-specific gene expression fingerprints in females we examined gene set 3: DE genes in common among ATR treatments at both 1 and 100 μg/L in females. Gene ontology (GO) analysis of

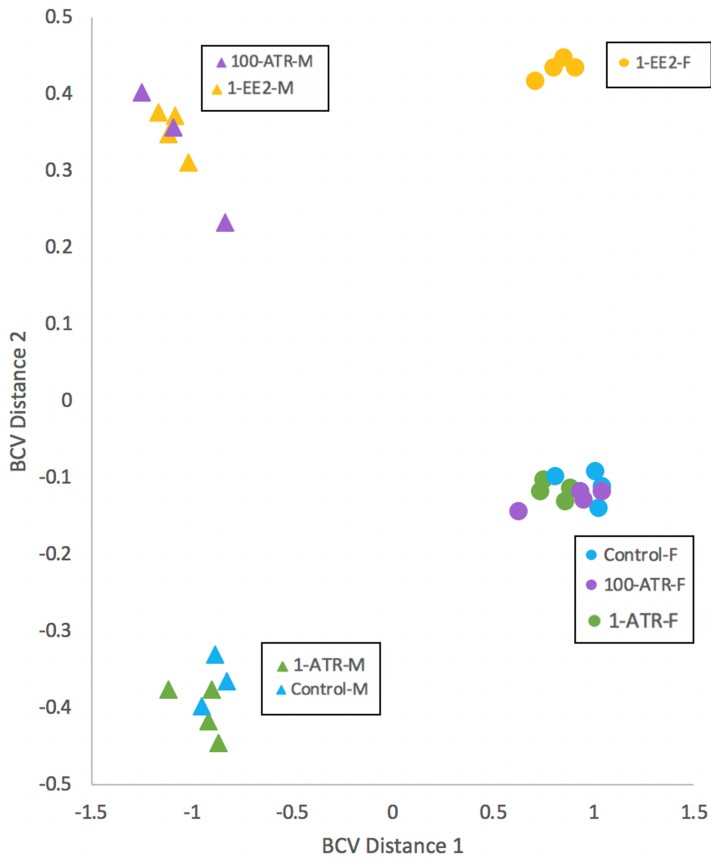

**Figure 3 Multidimensional scaling (MDS) with normalized counts.** MDS of transcript expression patterns in juvenile largemouth bass gonads in which the distance between individuals is based on biological coefficient of variation (BCV). Individual samples are represented by triangles (males) or circles (females). 1-ATR, 1 µg/L Atrazine; 100-ATR, 100 µg/L Atrazine; 1-EE2, 1 ng/L 17α-ethinyles-tradiol.

differentially expressed annotated transcripts (based on *Danio rerio* annotation) identified several significantly overrepresented terms among the three gene sets of interest tested. Among DE genes responsive to both ATR and EE2 treatments in males, there was an overall trend of down-regulation of female-biased genes, contrary to our prediction of a trend for upregulation of genes in this set (Table 2). Two GO terms that were enriched among DE gene sets from males of all treatment groups were ribosome biogenesis and small molecule metabolic process (Table 2). Common GO terms among male and female groups were transport and transmembrane transport. Whereas the male response was typically down-regulation of female-biased genes, the female response was typically upregulation of genes in overrepresented GO terms (Table 2).

### Comparative meta-analysis

The comparative meta-analysis identified specific genes of interest that have been shown to respond to both ATR and EE2 exposure (File S1). This analysis also provided a summary of literature in which those contaminant gene interactions could be found. Of our identified genes of interest, *cytochrome P450, family 11, subfamily A, polypeptide 2*

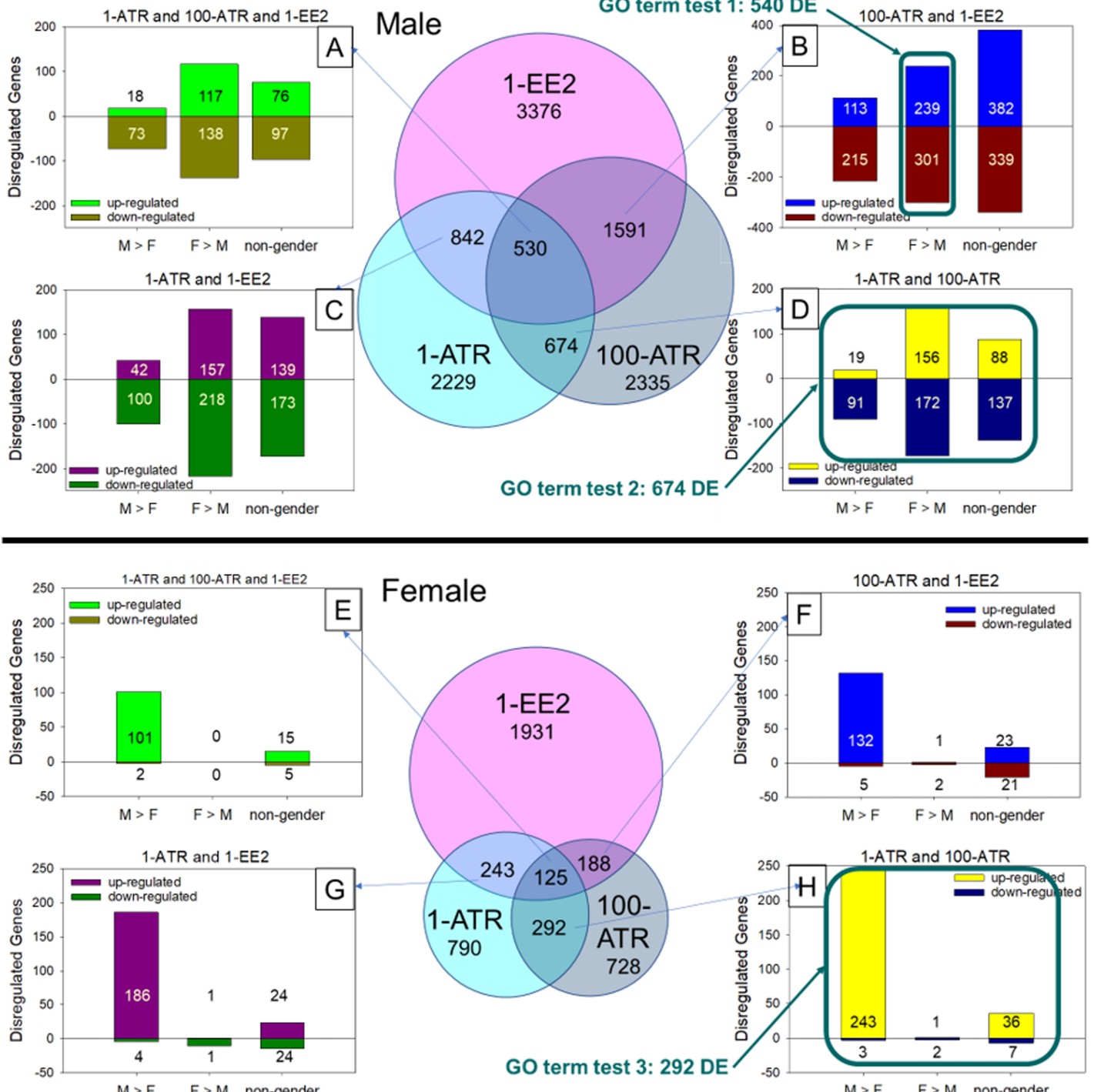

**Figure 4 Comparison of differentially expressed transcripts in each test group by sex.** Diagram of overlapping sets of differentially expressed (DE) transcripts that were common between sample groups and treatments. M, male; F, female; F > M, female-biased transcripts that had significantly greater expression in females than males in the controls; M > F, male-biased transcripts that had significantly greater expression in males than females in the controls; 1-ATR, 1 µg/L Atrazine treatment; 100-ATR, 100 µg/L Atrazine treatment; 1-EE2, 1 ng/L 17α-ethinylestradiol treatment. DE genes that did not have uniform regulation responses between treatments were not represented in the number of up- and down-regulated genes in bar graphs (A)–(H), but were included in the total number of DE genes in the Venn diagrams. There are 28,314 total DE transcripts. Conditions used to frame the three gene ontology (GO) queries are boxed.

**Table 2 Overrepresented GO terms for gene sets of interest.**

| Gene sets | Category | p-value | #DE genes | #Genes in category | GO term | #Genes down-regulated | #Genes up-regulated |
|---|---|---|---|---|---|---|---|
| 1: female specific transcripts altered in both the 1 ng/L EE2 and 100 µg/L Atrazine treatments | GO:0006605 | 0.0033 | 6 | 96 | Protein targeting | 6 | |
| | GO:0042254 | 0.00757 | 6 | 127 | Ribosome biogenesis[a] | 6 | |
| | GO:0007005 | 0.00975 | 5 | 89 | Mitochondrion organization | 5 | |
| | GO:0006913 | 0.0149 | 4 | 63 | Nucleocytoplasmic transport | 4 | |
| | GO:0006810 | 0.038 | 29 | 1,352 | Transport[b] | 16 | 13 |
| | GO:0044281 | 0.0473 | 14 | 564 | Small molecule metabolic process[c] | 12 | 2 |
| 2: transcripts altered in males in both the 1 and 100 µg/L Atrazine treatments | GO:0006412 | 0.00131 | 14 | 282 | Translation | 14 | |
| | GO:0044281 | 0.00329 | 23 | 564 | Small molecule metabolic process[c] | 21 | 2 |
| | GO:0006520 | 0.00336 | 9 | 133 | Cellular amino acid metabolic process | 8 | 1 |
| | GO:0055085 | 0.00382 | 22 | 495 | Transmembrane transport[d] | 15 | 7 |
| | GO:0009058 | 0.0262 | 54 | 1,916 | Biosynthetic process | | |
| | GO:0042254 | 0.0396 | 6 | 127 | Ribosome biogenesis[a] | 6 | |
| | GO:0008150 | 0.0464 | 203 | 8,605 | Biological process | | |
| 3: transcripts altered in females in both the 1 and 100 µg/L Atrazine treatments | GO:0006810 | 0.00196 | 18 | 1,352 | Transport[b] | 1 | 17 |
| | GO:0007267 | 0.00211 | 6 | 249 | Cell–Cell signaling | | 6 |
| | GO:0055085 | 0.00708 | 9 | 495 | Transmembrane transport[d] | | 9 |
| | GO:0007010 | 0.0189 | 7 | 370 | Cytoskeleton organization | | 7 |
| | GO:0007165 | 0.026 | 19 | 1,709 | Signal transduction | 1 | 18 |
| | GO:0002376 | 0.0271 | 7 | 428 | Immune system process | | 7 |
| | GO:0051604 | 0.0345 | 2 | 42 | Protein maturation | | 2 |

**Note:**
Gene ontology analysis of differentially expressed (DE) annotated transcripts identified by gene set 1, 2, or 3. Only GO terms that were significantly overrepresented ($p$-value < 0.05) are presented. Corresponding letters identify common GO terms between gene sets.

(cyp11a2), steroidogenic acute regulatory protein (star), cytochrome P450, family 1, subfamily A (cyp1a), DEAD (Asp-Glu-Ala-Asp) box polypeptide 4 (ddx4 (previously vasa)), SAM domain and HD domain 1 (samhd1) and wingless-type MMTV integration site family, member 5b (wnt5b) showed differential expression from the control group in at least one treatment group in males (Fig. 5).

## DISCUSSION

In this study we examined the potential endocrine disrupting effects of a common use herbicide (ATR) and a model estrogen (EE2) in developing largemouth bass. We evaluated both somatic growth and gonad development, as well as global gene expression in isolated developing gonad tissue. Our observation of near-complete sex reversal in the 10 ng EE2/L treatment confirmed that early development was a sensitive window for estrogenic endocrine disruption in LMB and our gene expression results were consistent with the hypothesis that ATR exposure induces some estrogenic responses in the developing gonad.

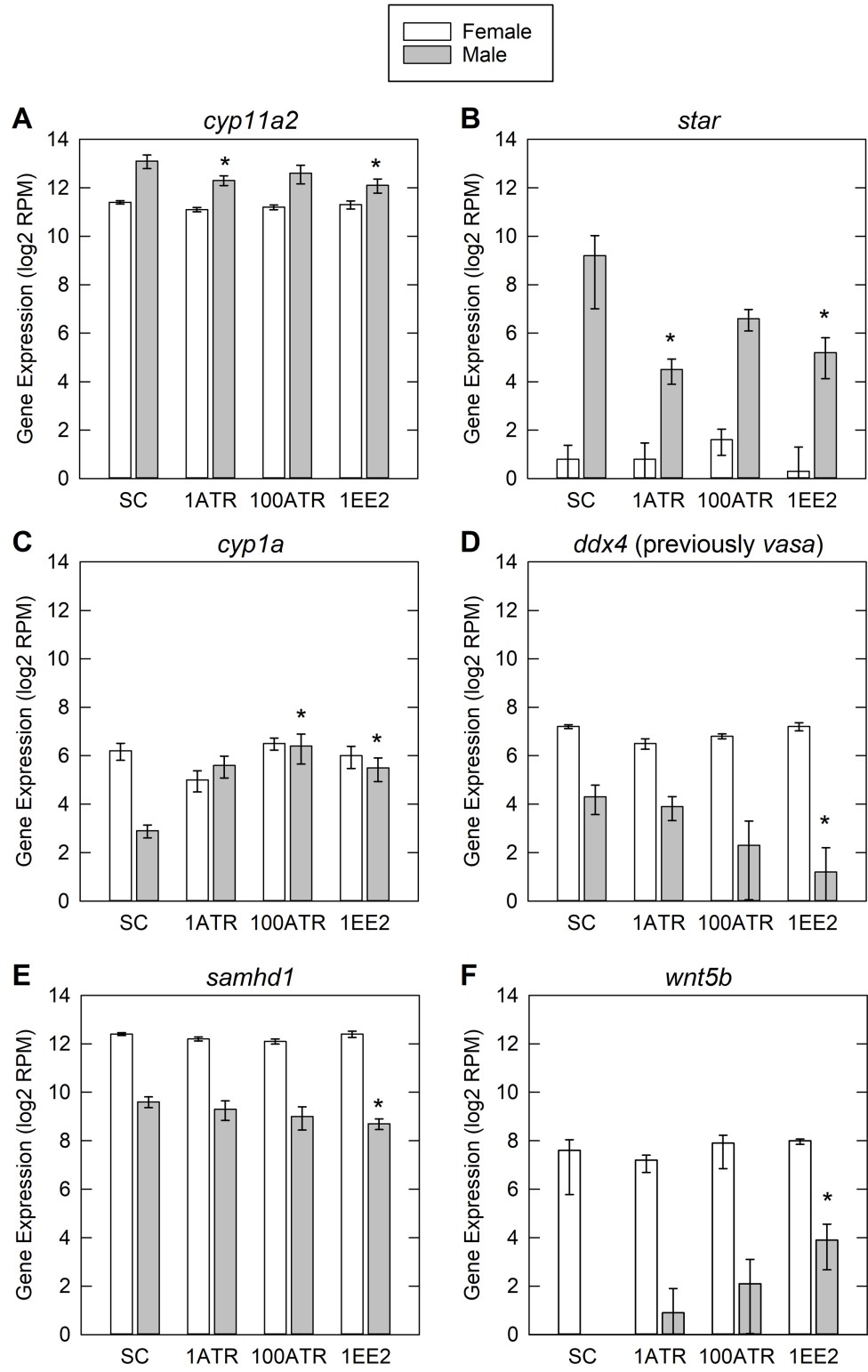

**Figure 5 Differential expression data for genes of interest.** Genes of interest were based on literature search and comparative meta-analysis. Genes presented here are those that were significantly altered in one or more treatment groups. SC, solvent control; 1-ATR, 1 μg/L Atrazine treatment; 100-ATR, 100 μg/L Atrazine treatment; 1-EE2, 1 ng/L 17α-ethinylestradiol treatment. Error bars are one standard error from the mean. The y-axis is in reads per million (RPM) on a log base 2 scale. Asterisk denotes

**Figure 5** (continued)
significant differential expression compared to SC. Gene symbols and names: (A) *cyp11a2, cytochrome P450, family 11, subfamily A, polypeptide 2*; (B) *star, steroidogenic acute regulatory protein*; (C) *cyp1a, cytochrome P450, family 1, subfamily A*; (D) *ddx4* (previously *vasa*), *DEAD (Asp-Glu-Ala-Asp) box polypeptide 4*; (E) *samhd1, SAM domain and HD domain 1*; (F) *wnt5b, wingless-type MMTV integration site family, member 5b*.                               

## Sex reversal, gonad growth and somatic growth

Overall, there was an absence of response to ATR exposure in growth, sex ratio and gonad morphology, while EE2 exposure elicited effects on all three. An increase in growth was observed in the EE2-treated female LMB. In the literature there have been both increased and decreased growth observed in fish exposed to EE2. However, low concentration exposures during development have been shown to act like a growth promoter (*Chen et al., 2017*; *Luzio et al., 2015*; *Örn et al., 2003*), as was observed in the current study. All but one fish (94/95) in the 10 ng EE2/L treatment were histologically identified as female.

Sex reversal resulting from exposure to high concentrations of EE2 has been observed in other species at exposure concentrations of 3 ng EE2/L and above (*Caldwell et al., 2008*). The LMB under the current exposure conditions appear to be as sensitive to sex reversal resulting from developmental EE2 exposure as the small fish model species zebrafish (*Hill & Janz, 2003*; *Weber, Hill & Janz, 2003*) and fathead minnow (*Van Aerle et al., 2002*). LMB appear to be more sensitive to sex reversing effects than what has been observed in three-spined stickleback (*Peters et al., 2010*), medaka (*Scholz & Gutzeit, 2000*) and sheepshead minnow (*Zillioux et al., 2001*). Previous studies have shown that exposure to a concentration of EE2 over a species-specific threshold for sex reversal throughout early development led to organizational changes, and although the sex-reversed phenotype reverted to a phenotype that matched genetic sex in some individuals after a recovery period during adulthood, reproductive dysfunction persisted (*Larsen, Bilberg & Baatrup, 2009*). In wild populations of LMB sex differentiation occurs in spring and summer, which coincides with storm events and typically greater exposures of EDCs in surface water, particularly in areas dominated by agricultural land use (*Gall et al., 2011*).

The mechanisms of sex differentiation in this species have not been characterized well. Since the window of sex differentiation is extremely sensitive to endocrine disruptors in all the vertebrate species tested so far, the EDC effects in LMB germ cells may lead to reproductive impairment later in life, which we did not examine. Such far-reaching effects would be detrimental for LMB populations in EDC-contaminated sites.

## Gene expression alterations
### Differential expression

The male LMB gonads were sensitive to estrogenic exposures, as observed by the apparent sex reversal in the 10 ng EE2/L treatment and the overall response in gene expression in the 1 ng EE2/L treatment. We had hypothesized upregulation of female-specific genes in testes among males exposed to EE2. The most differentially expressed genes were those more highly expressed in females, however, most DE genes were downregulated.
This counterintuitive result may be caused by negative feedback, leading to downregulation of the female-biased genes in the 1 ng EE2/L treatment.

During this period of gonad development, sex differentiation is just beginning in the males, so effects on gene expression patterns during this stage could lead to alterations in later gonad morphology and possibly function. The earlier stage of gonad differentiation in males compared to females may possibly account for some of the greater responsiveness of gene expression in males, as the fish may be more resistant to the influence of exogenous hormones as gonads become more developed (*Blázqueza et al., 1995*). The relationship between the observed sex reversal and gene expression changes in males to the formation of intersex in adults was not tested in this study. Due to the undifferentiated testes at the sampling time point intersex was not able to be identified. No morphological abnormalities were observed in either treated or control fish during the histological analysis and sex identification.

Liver gene expression of *vitellogenin 1* (*vtg1*), a commonly assayed biomarker of estrogen exposure in fish (*Bowman & Denslow, 1999*), was not measured in this study, and expression was not observed in gonad. We observed sex-specific expression of three zona pellucida genes, *zona pellucida glycoprotein 3a, tandem duplicate 2* (*zp3a.2*), *zona pellucida glycoprotein 3b* (*zp3b*) and *zona pellucida glycoprotein 3d tandem duplicate 2* (*zp3d.2*), which are generally estrogen-regulated and expressed in ovary and/or liver in adults (*Onichtchouk et al., 2003*). Expression of the three zp genes was female-biased, but there were no significant effects of treatment on expression.

### Gene Ontology analysis

Many of the GO terms overrepresented in this analysis were, in general, involved with protein synthesis and transport. Those terms that were found to be common among the sexes had responses in opposite directions, with differentially expressed genes in those categories being largely downregulated in males and upregulated in females exposed to ATR. Downregulation of these processes in developing males may lead to effects later in gonad development.

A GO term of particular interest that was overrepresented for transcripts altered in females in both 1 and 100 μg/L ATR treatments was immune system process. All of the DE genes were upregulated in this term. ATR has been shown to alter immune function in juvenile fish (*Kreutz et al., 2012*). Estrogen is also known to interfere with immune function in fish (*Burgos-Aceves et al., 2016*). There are still gaps in our knowledge of the potential effects of these contaminants on juvenile LMB and potential effects on disease susceptibility. There is even less known about how contaminants of interest act in real world mixtures and what role they potentially play in concert to modulate the immune function and disease-resistance of wild populations.

### Comparative meta-analysis

The annotation used for the analysis was *Danio rerio*, as this is the taxonomically closest species to LMB that has an annotated genome. There are, and will continue to be, limitations in this kind of study until the LMB genome can be fully sequenced and

annotated. A reciprocal best-match criterion between *D. rerio* and LMB transcripts was used to strengthen annotation inferences, which reduced the number of gene proxies to 11,916 out of approximately 25,549 coding genes in *D. rerio*. The number of genes in LMB is unknown, but the N50 of the assembly was not low by transcriptomic standards (3,545 bp) and only one tissue was analyzed. Nonetheless, partial capture of coding sequences, retention of non-coding sequence and alternative splicing remain challenges to annotation. We did not attempt to assess alternative splicing with these data, as robust identification of splice variants requires a genomic reference.

In developing fish, genes involved in steroidogenesis and gonad development have been shown to either be upregulated or downregulated with EDC exposure depending on the timing, duration, and concentration of the exposure (*Leet, Gall & Sepulveda, 2011*). In the current study there was a general trend of downregulation of DE genes in males, including *cyp11a2*, *star* and *ddx4* (Fig. 5), possibly indicating a negative feedback from EE2 and low concentrations of ATR (*Baron et al., 2005*; *Filby et al., 2007*; *Leet et al., 2015*). The genes of interest identified by the comparative meta-analysis are involved in steroidogenesis, metabolism, and gonad development. The gene products of *star* and *cyp11a2* are required for production of all steroid hormones; star encodes a cholesterol transporter which is rate-limiting for steroidogenesis, and *cyp11a2* encodes a cholesterol side-chain cleavage enzyme (*Arukwe, 2008*). Downregulation of *star* has been observed in adult human granulosa cells, mediated by an atrazine-induced increase in phosphodiesterase activity (*Pogrmic-Majkic et al., 2018*). The enzyme encoded by *cyp1a* is involved in metabolic breakdown of xenobiotics and steroids (*Otte et al., 2017*). The RNA helicase encoded by *ddx4* (previously *vasa*) is a regulator of translation and is required for primordial germ cell migration (*Li et al., 2009*). *Samhd*, an immune-related gene, was also downregulated in the current study. In contrast, a previous study in early life stage zebrafish showed *samhd* to be upregulated in response to exposure to ATR (*Weber et al., 2013*). *Wnt5b* is an extracellular signaling molecule and morphogen involved in cell differentiation and formation and maintenance of tissues and organs (*Yang, 2012*).

However, a few of the DE genes of interest were upregulated in males. *Wnt5b* was upregulated in the EE2 treatment, and *Cyp1a* was also upregulated in EE2 and high ATR treatment males. *Cyp1a* has previously been shown to be upregulated in fish where ATR exposure lead to DNA strand breaks and damaged blood cells (*Chang et al., 2005*).

## CONCLUSION

The early stage in LMB gonad development assessed in the current study was seen to be sensitive to molecular responses to EDCs, and organizational alterations in the form of sex reversal with exposure to a high concentration of a potent estrogen. Exposure to ATR resulted in changes in gene expression, with both similarities and differences compared to pathways activated by estrogen. Exposure to the strong estrogenic EDC EE2 set developing fish on a path to physiological change that was observed across multiple levels of biological organization: changes in gene expression, changes in histology, and changes in morphology. Thus, our results delineate pathways from estrogenic exposures to adverse outcomes in a major sport fish. The role of ATR in population impairments

observed in the field remains unclear. It is possible that the combined impacts of mixtures of contaminants including atrazine lead to population-level effects on reproduction and disease resistance (*Berninger et al., 2019*). Investigation of effects of mixtures at multiple levels of biological organization may help reveal diagnostic biomarkers of pathways leading to adverse population-level effects. In wild populations early sex differentiation occurs in the spring and summer, which coincides with storm events and typically higher concentrations of EDCs in surface water, particularly in areas dominated by agricultural land use. To our knowledge this is the first examination of developmental and molecular responses to EDCs in juvenile LMB. This study can serve as an initial piece of the larger picture of the sensitivity of developing bass to contaminants of interest in the Chesapeake Bay Watershed. Additional studies are being conducted in adult LMB exposed to mixtures of contaminants of interest. Controlled laboratory exposures with field-relevant sport fish can provide a basis for identification of specific mechanisms of action, determination of effect concentrations, and establishment of cause and effect linkages for contaminants and other stressors that may limit the health and growth of wild populations.

## ACKNOWLEDGEMENTS

We are grateful for the assistance of R. Claunch, J. Candrl and V. Velez (Columbia Environmental Research Center, US Geological Survey, Columbia, MO, USA). Any use of trade, product, or firm names is for descriptive purposes only and does not imply endorsement by the US government.

### Funding

The present study was supported by the US Geological Survey, Contaminants Biology Program, Environmental Health Mission Area. The funders had no role in study design, data collection and analysis, decision to publish, or preparation of the manuscript.

### Grant Disclosures

The following grant information was disclosed by the authors:
US Geological Survey, Contaminants Biology Program, Environmental Health Mission Area.

### Competing Interests

The authors declare that they have no competing interests.

### Author Contributions

- Jessica K. Leet conceived and designed the experiments, performed the experiments, analyzed the data, prepared figures and/or tables, authored or reviewed drafts of the paper, and approved the final draft.

 

- Catherine A. Richter conceived and designed the experiments, analyzed the data, prepared figures and/or tables, authored or reviewed drafts of the paper, and approved the final draft.
- Robert S. Cornman analyzed the data, prepared figures and/or tables, authored or reviewed drafts of the paper, and approved the final draft.
- Jason P. Berninger analyzed the data, prepared figures and/or tables, authored or reviewed drafts of the paper, and approved the final draft.
- Ramji K. Bhandari conceived and designed the experiments, authored or reviewed drafts of the paper, and approved the final draft.
- Diane K. Nicks conceived and designed the experiments, performed the experiments, analyzed the data, prepared figures and/or tables, and approved the final draft.
- James L. Zajicek conceived and designed the experiments, authored or reviewed drafts of the paper, and approved the final draft.
- Vicki S. Blazer analyzed the data, authored or reviewed drafts of the paper, and approved the final draft.
- Donald E. Tillitt conceived and designed the experiments, authored or reviewed drafts of the paper, and approved the final draft.

## Animal Ethics

The following information was supplied relating to ethical approvals (i.e., approving body and any reference numbers):

The U.S. Geological Survey Columbia Environmental Research Center Institutional Animal Care and Use Committee (CERC IACUC, Columbia, MO, USA) approved this research.

## Data Availability

The RNAseq data are available at NCBI BioProject: PRJNA485177.

Other relevant data are within the article and the Supplemental Files.

Raw data can be found through the US Geological Survey and is publicly available at DOI 10.5066/P93ZE9D6.

## Supplemental Information

Supplemental information for this article can be found online at http://dx.doi.org/10.7717/peerj.9614#supplemental-information.

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
