# Peer review of "Effects of early life stage exposure of largemouth bass to atrazine or a model estrogen (17α-ethinylestradiol)"

_PeerJ, doi:10.7717/peerj.9614_

## Round 0.1 · original submission · Minor Revisions

The reviewers have some good comments on your manuscript and were overall very complementary. Please use their suggestions and revise your manuscript as appropriate.

Reviewer 1 ·

Basic reporting

The manuscript is generally well written and professional, but there are some awkward phrasings throughout the manuscript that could be revised. The best example is the introduction of intersex condition on lines 53-59. The discussion of intersex could also be tempered considering the inability of the study to identify developed testes in this age of LMB and therefore intersex gonads in LMB.

The cited references are appropriate for species; however, one topic that would add to the manuscript would be to include a discussion of proposed mechanisms of action for ATR. There are manuscripts that suggest that ATR does not interact with the classical estrogen receptor, which would need to be mentioned due to the comparison with EE2. Also, there is research to suggest that ATR changes the expression of Star in rat granulosa cells, which should be mentioned due to the similarity in results in this manuscript. The lack of mention of prior mechanistic work is the greatest weakness of the manuscript but would be relatively simple to address.

The manuscript is well organized and raw data is shared in the supplemental files. In general, the figures are relevant and high quality; however, there are a couple of notes and suggestions that might be helpful.
1. Figure 1 – No comment
2. Figure 2 – The legend could be reworked to include the magnification during the description. For example “Example of gonad development at 17 dps at 100x (A) and 400x (B) magnification, 33 dps at 40x (C) and 400x (D) magnification…”. I think this would help clarify the images. Also, the legend should mention which group the images are from. Are these control fish or exposed fish? Finally, image F is a little dark and out of focus. Is there a replacement image that could be used, especially considering it doesn’t seem to match the boxed area in E.
3. Figure 3 – BVC could be defined in the legend to help explain the axis titles.
4. Figure 4 – This figure is busy, but very informative. I think it does a good job of presenting a lot of data.
5. Figure 5 – I think this figure could be removed because it seems to not add additional information compared to figure 4.
6. Figure 6 – The error bars need to be defined. Also, some error bars are very long on the bottom compared to the top and this may need to be corrected. I would also suggest defining the gene names in the figure legend. RPM on the Y-axis needs to be defined in the figure legend as well.
7. Both table 1 and 2 look fine.

The manuscript is self-contained and has relevant results.

Experimental design

The research appears to be original primary research within the aims and scope of the journal. The research questions are reasonably well defined but are relevant and meaningful. The manuscript does address a gap in the literature (effects of developmental exposure of LMB to ATR and EE2), and nicely defends the relevance of the study in the introduction and discussion, even if the claim of identifying molecular initiating events is a slight overstatement. It would be more appropriate to say ‘pathways’ rather than ‘molecular initiating events’.

With respect to the materials and methods, overall, methods are well described and detailed. A few specific comments are given below:
1. Although it is common in the literature to only include a solvent control, the inclusion of a solvent free control (normal tank water) in addition to the solvent control is the best practice.
2. Were all 4 replicates from the same original batch of fry (tank replicates), or were they from different batches of fry (true biological replicates)?
3. More details into the types of tanks and size of tanks would be helpful.
4. Some of the details of methods are split over two sections. For example, it is unclear how samples were preserved for histologic analysis (lines 193-194), until the following section (205-206). It would be more clear to explain the details the first time the method is mentioned and revise the second mention to not repeat information.
5. Which samples were fixed in Z-fix and which were preserved in PAXgene tissue fix? Z-fix contains formaldehyde and PAXgene advertises that it does not. The different fixation methods will have different “normal” artifacts which could possibly influence histologic interpretation.
6. How was the p-value for differential expressed genes calculated (line 279)?
7. Why was no minimum fold-change threshold used (lines 281-282)?

Validity of the findings

The data and findings are sufficient for the journal standards and appear to be robust and statistically sound (minus the lack of method mentioned for differentially expressed genes). Overall, there are a few over statements (lines 98-99, 392-394), that should be tempered or clarified.

The discussion overall is generally well supported by the cited references (though a citation would be useful around lines 418-421), but could be expanded by the addition of a discussion of ATR related gene expression changes in other species or models. It would be important to note of some of the genes were differentially expressed in other species after ATR exposure.

One more thing to note is that ATR, like many EDCs, has been associated with non-monotonic dose responses. Important genes/pathways may have been missed due to the requirement of having DE in both 1 and 100 ug/L. Separate analyses may provide even more information.

A discussion of the significance of star, cyp11a2, vasa, wnt5b, cyp1a, and samhd1 would also be helpful to include. What are the full names of these genes and what are their functions?

The discussion of differentially expressed genes (lines 4432-436) raised some good points and could be expanded. Additionally, it should be made clear that intersex gonads could not be identified due to the developmental age of the LMB and lack of testicular development.

Additional comments

The manuscript has some areas that could be improved (see other sections), but overall is a nice, well-contained manuscript.

Reviewer 2 ·

Basic reporting

This manuscript by Leet and coauthors analyzes the effects of exposure to atrazine and ethinylestradiol on largemouth bass early life stages. The authors aim to understand the potential contribution of endocrine disrupting compounds on bass reproductive dysfunction. The paper is very well written and easy to read and provides novel information that could be useful for the management of bass species. The exposures performed here are extremely labor and time intensive and will definitely provide valuable information to understand endocrine-disrupting effects on bass populations. I do have a few minor comments that might help clarifying the paper.
- Please add locations at least state, to the areas of interest when possible
- Please add reference/link to the zebrafish genome version
- Line 371: Danio rerio
- Please add city, state, to the companies used (ie Qiagen, Illumina, etc)

Experimental design

- How many replicates were used for RNAseq? Line 114 mentions 4 fish and line 271 mentions 5
- Please add number reads/sample on the RNAseq results
- What was the quality of the RNA used?
- What was the rationale for not using a minimum fold change threshold?
- Was there any attempt to blast the sequences against largemouth bass sequences from the database, or was it considered that might not provide good enough coverage?
- Figure 5: please add more details to color legend
- While the gonads might still be undifferentiated we are likely talking about two quite different tissues (male and female gonads). What was the rationale behind analyzing them together?

Validity of the findings

no additional comment

Additional comments

This paper provides very valuable information on the potential adverse effects of endocrine disruptors to bass populations, particularly during the window of sex differentiation.

Reviewer 3 ·

Basic reporting

Overall, this is a well-written, interesting manuscript that clearly lays out a gap in understanding on this topic, defines objectives, outlines results, and puts into context the implications of the results. There are a few sections that may be clarified to aid in reader comprehension, and a few grammatical errors that may be easily fixed with a thorough proof-reading.

For example:
Line 53: "A high prevalence... have been observed..." should say "...has been observed..." to match the singular subject, “prevalence.”

Paragraph beginning on Line 73: Although this paper is clearly about fish, consider clarifying which taxonomic groups you are referring to in the topic sentence, since reproductive dysfunction and EDC exposure are issues in other taxa as well.

Lines 82-84: "There have been many studies evaluating the effects of EE2 on various fish species at different life stages... However, the exact mechanism of this effect has not been identified..."
To what effect, specifically, are you referring for which the mechanism has not been identified?
Or, did you mean to say "...mechanisms of these effects have not been identified..."?

Lines 94-95: Consider replacing “…changes in gene regulation…” to “…differences in gene regulation…” as the way you have it worded could imply to the reader that multiple time-points were sampled for gene regulation per treatment group.

Line 420: I think “far-reaching” should be hyphenated since it is used as an adjective.

Lines 470-472: The sentence beginning with “Samhd…” should be re-worded for clarity.

Lines 484-485: The sentence beginning with “From this analysis…” is unclear. Consider re-wording.

In the discussion, there are some sentences in which it may be initially unclear whether the authors are referring to previous studies or the current study. Consider adding some clarification.

Experimental design

Was vitellogenin expression (either protein or transcripts) analyzed in these fish? If so, this should be discussed in the manuscript. This is a commonly used biomarker of estrogen exposure in fish and would be valuable information to assess the effects of the EE2 exposure, as well as the potential estrogen-like effects of atrazine.

What do we know about the history of this LMB population? For example, is there any history of intersex, or estrogen exposure of predecessors? How long has the population been sustaining itself without re-stocking? (i.e. How genetically diverse/inbred are these fish?). Since there are genetic/epigenetic factors to consider, I think a short background about the site/population would be helpful. Consider adding a sentence or two to help the reader understand the population a bit better.

Approximately how many clutches of eggs were collected? Do you suspect that the eggs came from multiple females? The manuscript states the eggs were “recently deposited.” What exactly does that mean? Also, were the eggs mixed together or randomized prior to stocking the exposure aquaria? Consider adding a sentence or two to clarify these questions for the reader.

Exposure and chemical analysis methods section needs some clarification. From what I can see, there is no mention of how EE2 solutions were prepared. This needs to be addressed. Also, I’m assuming (since controls were 0.0001% ethanol) that ATR and EE2 was initially dissolved in ethanol? Please clarify.

Lines 143-146: This sentence is a bit unclear. Were fish removed during chloramine treatments? Were tanks treated with 10ppm chloramine continually after the first four weeks, or still just twice a week?

Validity of the findings

Although methods are mostly clearly described, there are important issues that should be addressed:
My biggest concern is with the histological methods used for sexing the juvenile fish. Presumed “male” gonads appear to be underdeveloped, and thus difficult, if not impossible to confidently identify the sex. In lines 327-329, the authors report only 70% agreement among blinded readers, which illustrates the potential for ambiguity. Methods used to identify females are pretty well-defined, and the representative histological images seem to show obvious ovarian tissue. However, identification of males was determined solely by the presence of germ cells and absence of primary oocytes (as stated in Lines 227-227), yet the only characteristic that differentiates a “male” from “inconclusive sex identification” appears to be the germ cells which may also be present in underdeveloped females, particularly if portions of the gonad were missed by histological sectioning. The overall area of gonadal tissue in the representative “male” histological images is very small (~50 x 100um, Fig. 2F). I am curious what criteria were used in evaluating these slides? Was there a certain amount of gonadal tissue area required for sex determination? How many germ cells were required in order to be considered a “male” vs. “inconclusive”? If only one germ cell was present, was that individual determined to be a male? In Lines 331-332, one individual in the high EE2 group is identified as a “presumptive male,” but elsewhere in the paper, males are not described as “presumptive.” Is there something unique about this individual other than its EE2 treatment, or are all of the “males” only presumed so? Please clarify. With my current understanding of these methods, I am not fully convinced by the determining characteristics of presumed male fish. These methods and definitions should be clarified for the reader, and the possibility of false identification of males should be discussed, since this could have an impact on the results and implications of the study.

---

## Round 0.2 · accepted · Accept

Thank you for your detailed responses to reviewer comments and for your willingness to revise your manuscript to make it more readable.